# Obstructive Sleep Apnoea in Children with Down Syndrome: A Multidisciplinary Approach

**DOI:** 10.3390/jpm13010071

**Published:** 2022-12-28

**Authors:** Melissa Borrelli, Adele Corcione, Roberto Rongo, Elena Cantone, Iris Scala, Dario Bruzzese, Stefano Martina, Pietro Strisciuglio, Ambrosina Michelotti, Francesca Santamaria

**Affiliations:** 1Department of Translational Medical Sciences, Pediatric Pulmonology, Federico II University, 80131 Naples, Italy; 2Department of Neurosciences, Reproductive Sciences and Odontostomatologic Sciences, School of Orthodontics, Federico II University, 80131 Naples, Italy; 3Department of Neurosciences, Reproductive Sciences and Ear Nose Throat Section, Federico II University, 80131 Naples, Italy; 4Department of Maternal and Child Health, Clinical Genetics, Federico II University, 80131 Naples, Italy; 5Department of Public Health, Federico II University, 80131 Naples, Italy; 6Department of Medicine, Surgery and Dentistry “Scuola Medica Salernitana”, University of Salerno, 84121 Salerno, Italy

**Keywords:** Down syndrome, obstructive sleep apnoea, sleep-disordered breathing, polygraphy, sleep questionnaire, ear, nose and throat examination, orthodontic evaluation

## Abstract

A comprehensive evaluation of obstructive sleep apnoea (OSA) may allow for the development of more efficient management of Down syndrome (DS). We aimed to evaluate the effect of a multidisciplinary approach to DS with OSA. A total of 48 DS children aged 4–12 years were prospectively investigated with nasal endoscopy, orthodontic examination, and overnight polygraphy (PG); the Italian Child Sleep Habits Questionnaire (CSHQ-IT) was filled out by the mothers. The total CSHQ-IT score was 63 (96% of children reporting sleep problems). The major ear, nose, and throat characteristics were enlarged palatine tonsils (62%), adenoid tonsils (85%), and chronic rhinosinusitis (85%). DS children showed orthognathic profile in 68% of cases, class I relationship in 63%, and cross-bite in 51%. PG revealed OSA in 67% of cases (37% mild, 63% moderate–severe). The oxygen desaturation index (ODI) was higher in the group with OSA (5.2) than with non-OSA (1.3; *p* < 0.001). The ODI was higher (*p* = 0.001) and SpO2 lower (*p* = 0.03) in children with moderate–severe OSA than with mild OSA. The apnoea–hypopnea index (AHI) and percentage time with SpO2 < 90% were higher in DS children with grade III than with grade I or II adenoids (5 vs. 1, *p* = 0.04, and 1.2 vs. 0.1, *p* = 0.01, respectively). No significant correlations were found between PG and the total CSHQ-IT score or orthodontic data. However, children showing associated cross-bite, grade III adenoids and size 3 or 4 palatine tonsils showed higher AHI and ODI than those without (*p* = 0.01 and *p* = 0.04, respectively). A coordinated multidisciplinary approach with overnight PG is a valuable tool when developing diagnostic protocols for OSA in DS.

## 1. Introduction

Down syndrome (DS), one of the most common causes of intellectual disability in childhood, is associated with relevant medical comorbidities, including congenital heart defects, thyroid dysfunction, and sleep-disordered breathing (SDB) [1]. The most frequent type of SDB reported in the DS population is obstructive sleep apnoea (OSA), while central sleep apnoea and nocturnal hypoventilation are less frequently reported [2].

The prevalence of OSA is estimated at 35–70% in DS compared with 1.2–5.7% of the general population [3,4]. Several abnormalities of the DS upper airways, along with generalized hypotonia, recurrent respiratory infections (RRIs) due to an immature immune system, and a propensity for obesity, favour the development of OSA [2]. Although many of the manifestations of OSA in DS are similar to those seen in non-DS individuals, the relative morbidity is significantly higher in DS. In fact, OSA can seriously impact the clinical course, and chronic hypoxia and progressive respiratory failure may eventually develop [5]. 

The high prevalence of OSA among the DS paediatric population has led to the recommendation of monitoring very soon the onset and progression of symptoms [6]. Questionnaires are frequently used to report the parental perception of sleep problems in the DS paediatric population [7,8]. However, the gold standard for screening OSA in DS children is polysomnography (PSG) as a poor correlation between parents’ reports and PSG results was reported [9,10]. Regrettably, not all hospitals have PSG available for the overnight registration of sleep events; moreover, PSG is often poorly tolerated by DS children. Finally, the high prevalence of DS-associated upper airway abnormalities underlines the need for monitoring upper airways and oral–dental disorders, which is known to significantly increase the risk of OSA in affected children [2].

In many complex chronic conditions due to genetic disorders like DS, a multidisciplinary approach is mandatory to prevent or cure the associated medical comorbidities [11]. Despite the relevant number of reports on OSA, no studies evaluating a multidisciplinary approach to the disorder in DS are available. This study prospectively compared the findings from overnight PG, parents’ sleep report, and ear, nose, and throat (ENT) and orthodontic evaluation with the aim of assessing a coordinated multidisciplinary approach to OSA in a cohort of children with DS.

## 2. Materials and Methods

A multidisciplinary team consisting of paediatric pulmonologists, a paediatric geneticist, a paediatric ENT specialist, and paediatric dentists from Federico II University designed a prospective observational study. A total of 48 children with DS were enrolled at the Department of Translational Medical Sciences, Federico II University, Naples, Italy. Children were eligible for inclusion if they had a confirmed diagnosis of DS by karyotype analysis and were 4–12 years old. This age range was constrained by the questionnaire we used to evaluate DS children’s and adolescents’ sleep habits and disturbances [8]. Exclusion criteria were upper and/or lower airway infections and asthma or wheezing exacerbations occurring in the past 4 weeks; previously diagnosed allergic asthma and/or rhinitis; evidence of any complex congenital or acquired heart disease associated with signs and/or symptoms of heart failure; history of very preterm birth (less than 32 weeks of pregnancy); need for chronic oxygen administration and/or invasive/non-invasive ventilation; treatment with inhaled bronchodilators and/or inhaled/systemic steroids in the previous 24 h or 2 weeks, respectively; use of anticonvulsant or psychoactive drugs that could affect sleep; and the presence of any active smoker parent. 

After enrolment, on the study day, all subjects’ mothers filled out the Child Sleep Habits Questionnaire [7] adapted to the Italian language (CSHQ-IT) [12]. On the morning of the study day, children underwent anthropometric measurements, ENT evaluation with nasal flexible fibreoptic endoscopy, and orthodontic evaluation. Children were grouped into weight categories based on age- and sex-adjusted body mass index (BMI) percentiles according to Centers for Disease Control and Prevention (CDC)classification (underweight, ≤5th percentile; normal weight, 5th–85th; overweight, 85th–95th; obese, ≥95th) [13]. On the night of the study day, a sleep study with polygraphy (PG) was performed. The study was approved by the Ethics Committee of Federico II University, Naples (protocol no. 104/19). Written informed consent was signed by the parents on behalf of their children. 

### 2.1. Parent Report of Sleep by the CSHQ-IT

As previously described, mothers who filled out the retrospective 33-item CSHQ-IT were asked to score children’s sleep behaviours over a ‘typical’ recent week [12] (Appendix A). Like the original CSHQ, the CSHQ-IT yields scores on eight subscales: bedtime resistance (6 items), sleep-onset delay (1 item), sleep anxiety (4 items), sleep duration (3 items), night waking (3 items), parasomnias (7 items), sleep-disordered breathing (3 items), and daytime sleepiness (8 items). Three additional questions concerned the evening bedtime, the morning wake-up, and the total sleep duration (including daytime sleep). Every item was scored on a 3-point scale: usually (5–7 times/week), sometimes (2–4 times/week), or rarely (0–1 time/week). A number of items on the questionnaire are reverse-scored so that higher scores consistently indicate problem behaviours. The total score was the sum of the responses obtained on each item with a range from 33 to 99, and the highest scores indicated the worst sleep habits. A cut-off score of 41 has been shown to accurately identify paediatric sleep disorders with sensitivity and specificity of 0.80 and 0.72, respectively [7]. 

### 2.2. ENT Examination

The ENT examination, performed in the wakefulness and sitting position by the paediatric ENT specialist, included the following:Palatine tonsillar hypertrophy evaluation. Tonsil size was graded from 1 to 4: size 1 if tonsils were hidden within the pillars, size 2 if tonsils extended to the pillars, size 3 if tonsils extended beyond the pillars but not to the midline (Figure 1a), and size 4 if tonsils extended to the midline [14].Nasal flexible fibreoptic endoscopy (Storz, Tuttlingen, Germany) without decongestant or local anaesthesia to evaluate the following:
(a)Adenoid tonsil hypertrophy. The adenoid tonsil size was graded into four degrees, i.e. Grade I when fibreoptic endoscopy imaging revealed adenoid tissue occupying only the upper segment of the nasopharyngeal cavity (≤25%) with almost completely free choanal openings; Grade II if the adenoid tissue was confined to the upper half part (≤50%) of the nasopharyngeal cavity with sufficiently pervious choana and perfect visualization of the tube ostium; Grade III if adenoid tissue occupied around 75% of the nasopharynx with partial involvement of the tube ostium and considerable obstruction of choanal openings; Grade IV if the adenoid tissue reached the lower choanal border without allowing visualization of the tube ostium (>75%) (Figure 1b) [15].(b)Presence or absence of nasal turbinate hypertrophy or septum deviation.(c)Nasal mucosa oedema, secretions, and polyps according to the Lund and Kennedy quantifying system [16]:
(c.1.)Nasal mucosal oedema: 0, absent; 1, mild/moderate; 2, polypoid degeneration.(c.2.)Secretion: 0, absent; 1, hyaline; 2, thickened and/or mucopurulent.(c.3.)Polyps: 0, absent; 1, restricted to the middle meatus; 2, extending to the nasal cavity.

The above-reported parameters were assessed bilaterally, with the total score (0–12) being the sum of each side score. A total score of >2 indicated chronic rhinosinusitis (CRS) [16].

### 2.3. Orthodontic Evaluation

An oral examination performed by a specialist in orthodontics (SM) included the evaluation of the following features: Facial profile, classified as convex, concave, or orthognathic [17].Molar relationship, classified on both the left and right sides, according to Angle’s classification as Class I, II, or III [18].Overbite (OVB), i.e. the extent of vertical overlap of the maxillary central incisors over the mandibular central incisors, which was measured by an intra-oral ruler. A final value of ≥4 mm and ≤0 mm indicated deep bite and open bite, respectively, while 0 < OVB < 4 was considered normal [18].Transversal molar discrepancy with skeletal contraction, recorded on both sides in centric relation, is classified as unilateral or bilateral posterior cross-bite [19].

Mothers were asked about sleep bruxism and oral parafunctional behaviours, the latter including awake bruxism, clenching, grinding, bit nailing, placing the tongue between teeth during speaking or swallowing, and thumb sucking [20].

### 2.4. Sleep Study


The overnight PG was performed using a cardiorespiratory device (Embletta MPR, Medcare Flaga, Reykjavík, Iceland) for a period of at least 5 h. Patients went to bed at a time of their preference, and studies were terminated when they awoke spontaneously in accordance with their home wake times. The following parameters were recorded: airflow through nasal pressure transducer, oxygen saturation by pulse oximetry (SpO2), pulse signals, thoracic and abdominal movements by inductance plethysmography, and body position. We manually scored PG according to the American Academy of Sleep Medicine Paediatric criteria [21]. Obstructive apnoea was defined as the presence of continued inspiratory effort associated with a >90% decrease in airflow for a duration of ≥2 breaths. Hypopnoea was defined as a ≥30% decrease in airflow for a duration of ≥2 breaths associated with a decrease in SpO2 by ≥3%. The PG results included the following:Apnoea–hypopnoea index (AHI), defined as the sum of all obstructive, central, and mixed apnoeas and hypopnoeas divided by hours of total sleep time.Oxygen desaturation index (ODI), defined as the number of times per sleep hour with SpO2 decrease ≥3%.Mean SpO2.Percentage of the total recording time spent with SpO2 below 90% (T90).


OSA was defined as AHI > 1, and further classified as mild OSA if AHI > 1 and ≤5, moderate OSA if AHI > 5 and <10, and severe OSA if AHI ≥ 10 [21].

### 2.5. Statistical Analysis

All statistical analyses were conducted using the Statistical Platform R (version 4.0.1; R Foundation for Statistical Computing, Vienna, Austria). Numerical variables were described either as mean ± SD or as median and interquartile range in cases of distribution showing substantial asymmetry. Accordingly, any assessment of the statistical significance of the differences between the two groups (e.g. OSA versus non-OSA group, mild OSA versus moderate–severe OSA group, children with grade III versus children with grade I or II adenoid tonsil hypertrophy) was based on Student’s *t*-tests or Mann–Whitney tests. Only for CSHQ-IT data, DS children were compared to sex-, BMI-, and age-matched children without DS, who were part of the study cohort in our previously published report [12]. Categorical variables were reported using frequencies and percentages, and associations between qualitative variables were assessed by the X2-test (or the Fisher exact test when appropriate). The Spearman correlation test was used for the correlation analysis of continuous variables. A *p*-value of < 0.05 was considered statistically significant. 

## 3. Results

The DS children’s mean age was 8 ± 2.5 years (27 females, 53% of the total). The mean BMI was 19.9 ± 3.7 kg/m^2^. Of all DS patients, 40% were overweight and 29% were obese, while the remaining 31% had normal weights. 

All mothers completed the CSHQ-IT. The mean CSHQ-IT total score was 63 ± 5.1. In a relevant proportion of cases (96%), the CSHQ-IT total score was >41. Table 1 summarizes the total and subscale scores of the CSHQ-IT completed by the mothers of DS children.

We compared the total and subscale scores of the CSHQ-IT completed by the mothers of DS children with those of 48 mothers of children without DS. We found that the total scores from DS and healthy children’s mothers were significantly different (63 ± 5.1 vs. 45.8 ± 6.9, *p* = 0.001; Appendix A). Of all subscale scores, only the “sleep onset delay” scores were not different (*p* = 0.93), while the remaining were significantly higher in the DS group than in the healthy children cohort (*p* < 0.05 for all subscales).

No patients had previously undergone adenoidectomy or tonsillectomy. Of the 48 children, 34 (71%) and 47 (98%) successfully underwent ENT and orthodontic evaluation, respectively, while in the remaining 14 (29%) and 1 (2%) cases, respectively, the procedures were not performed because of lack of collaboration.

Table 2 summarizes the results of the ENT assessment. Twenty-one patients (62%) had a tonsil size ≥3. Also, 85% of DS children had adenoid hypertrophy, with 14 (41%) and 15 (44%) cases showing grade II and III hypertrophy, respectively. Eight DS children had both grade III adenoid hypertrophy and tonsil size ≥3. Most children (85%) had endoscopic evidence of CRS, as indicated by the Lund–Kennedy total score >2. Nasal turbinate hypertrophy and deviated septum were detected in 23 (68%) and 17 (50%) DS children, respectively, and 18 (53%) presented both features. 

Orthodontic results are summarized in Table 3. The most prevalent facial profile (32 cases; 68%) was orthognathic, while concave and convex profiles were present in 10 (21%) and 5 cases (11%), respectively. The molar relationship was evaluated in 38/47 patients because this evaluation can be performed only after the first molar appearance. Of all children, 24 (63%) showed Class I molar relationship, while the remaining 21% and 16% had Class III or Class II molar relationship, respectively. Normal OVB was observed in 81% of DS children; among children with OVB not in the normal range, open bite (17%) were more frequent than deep bite (2%). A total of 24 patients (51%) showed posterior cross-bite. In 47% and 30% of cases, bruxism and other oral parafunctional behaviours were reported, respectively. 

All DS children enrolled were able to complete the PG, and all studies were interpretable. The findings of PG are summarized in Table 4. The median total sleep time was 480 minutes. The median AHI was 2.2. According to AHI values, 67% of the patients (32 cases) had OSA, 20 (62%) had mild OSA, and 12 (37%) had moderate–severe OSA. The median of SpO2 mean was 96.3%; median ODI and T90 were 4 and 0.2%, respectively. 

When the DS study cohort was divided into the OSA (AHI > 1) and non-OSA group (AHI ≤ 1), no statistically significant differences were found in the CSHQ-IT total and subscale scores (Appendix A), and in the findings from both the ENT (Appendix A) and the orthodontic evaluation (Appendix A). Within the OSA group, we did not find any statistically significant difference in the CSHQ-IT scores as well as ENT and orthodontic findings of DS children with mild OSA compared to DS children with moderate–severe OSA (Appendix A). As shown in Figure 2, the evaluation of oximetry indices showed that ODI was significantly higher in the OSA group than in the non-OSA group [5.2 (2.6; 9.6) vs. 1.3 (0.7; 2.4), *p* < 0.001] (Figure 2a). Mean SpO2 and T90 were not statistically significantly different between the two groups (Figure 2b,c). In the moderate–severe OSA group, the ODI was significantly higher while mean SpO2 was significantly lower than in the mild OSA group [ODI 9.6 (7.4; 12.5) vs. 4.4 (1.9; 5.7), *p* = 0.001; mean SpO2 95% (93; 96) vs. 96% (95; 97), *p* = 0.03] (Figure 2a,b). While T90 was higher in the group of DS children with moderate–severe OSA than in DS children with mild OSA, there were no statistically significant differences [T90 1.5 (0.1; 16.4) vs. 0.2 (0; 1.9), *p* = 0.05] (Figure 2c). 

We compared the PG parameters with the demographics of DS children and found that only the mean SpO2 was negatively related to age (r = −0.35, *p* = 0.01) and BMI (r = −0.50, *p* = 0.001).

Table 5 summarizes the correlation between all CSHQ-IT total and subscale scores and PG variables. No PG parameters correlated significantly with the total CSHQ-IT score. When we evaluated the different subscales of the CSHQ-IT, we found an inverse correlation between the score of the subscale exploring “sleep-disordered breathing” and mean SpO2 (r = −0.35; *p* = 0.016). A direct correlation was shown between the score of the subscale exploring “sleep duration” and the percentage of the total recording time spent with SpO2 below 90% (r = 0.33; *p* = 0.022).

When we looked at the relationship between overnight PG data and ENT findings, AHI and T90 values were significantly higher in DS children with grade III than in those with grade I or II adenoid tonsil hypertrophy (*p* = 0.04 and *p* = 0.01, respectively), while no significant associations were found between PG parameters and other ENT variables (Table 6). 

No significant associations were found between PG parameters and orthodontic data. However, DS children who had associated unilateral or bilateral posterior cross-bite, grade III adenoid tonsil hypertrophy, and size 3 or 4 palatine tonsil hypertrophy showed higher values of AHI and ODI than patients without these associated findings (Table 7). 

## 4. Discussion

In this cross-sectional analysis of a cohort of children with DS, we found that 67% of cases had OSA and that ENT and orthodontic manifestations were highly prevalent. The novel finding of the study was provided by the comparison of data from overnight sleep PG to parents’ report of sleep and to ENT and dentist assessment. We found that the parents’ report on their kids’ sleep was inaccurate. Finally, even by dividing the study population into children with mild or moderate–severe OSA and children without OSA, we did not find any statistically significant difference in CSHQ-IT total or subscale scores or in either the ENT or orthodontic findings. Only nocturnal oximetry parameters appeared to be effective for identifying children with DS at risk of OSA.

The DS respiratory phenotype is dominated by OSA, an entity with complex and different aetiology. Many factors predispose DS patients to OSA, although not all have the same impact on the individual clinical course [22]. Children with DS are born with multiple anomalies in the craniofacial and ENT systems. These, combined with tonsillar and adenoid enlargement that may gradually develop at preschool age, greatly reduce the oral cavity. Generalized pharyngeal muscle hypotonia induces collapse of the upper airway during sleep, further worsened by overweight or obesity [2]. Finally, immune system dysfunction increases the susceptibility to respiratory infections [23], leading to upper airway obstruction and OSA, which, in turn, favour the persistence of RRI and their early development [10]. For all the above, a paediatric multidisciplinary team including pulmonologists, ENT specialists, and dentists who join forces with DS healthcare providers is mandatory.

In the current study, we performed a multidisciplinary approach to OSA in a cohort of DS children. Data from overnight sleep PG were compared to parents’ reports of the Italian version of the CSHQ and to the ENT and dentist assessment. We found that mothers of DS children reported sleep problems in 96% of our study children. In DS children, the total and the subscales scores, except for the sleep-onset delay, were significantly higher than in typically developing subjects, suggesting that DS children were much more likely than non-DS peers to have frankly disturbed sleep. These results are consistent with previous literature about DS parents’ reports on their kids’ sleep habits [24,25].

Major current ENT findings were enlarged palatine tonsils (62%), adenoid tonsils (85%), and endoscopic evidence of chronic rhinosinusitis (85%), confirming that ENT manifestations are highly prevalent in DS [26]. 

In addition to morphologic abnormalities of the nose, ear canals, and Eustachian tube [10], enlarged adenoids closer to the Eustachian tube and medially positioned hypertrophic palatine tonsils further reduce the nasopharynx and oropharynx spaces. Chronic otitis media with effusion and hearing loss could occur early. All the conditions described above explain why ENT consultations are frequently required in DS [26]. 

Orthodontic evaluation revealed that more than half of our DS cohort showed posterior cross-bite. This value is much higher than the 12% prevalence of posterior cross-bite reported in non-DS children [18]. We also found that 17% and 21% of current DS children had open bite and class III molar relationship, respectively, as previously reported by other DS studies [27,28]. On the other hand, recent studies have reported a lower prevalence of either Class III or open bite in Italian non-syndromic schoolchildren [18,29]. Malocclusion in DS results from craniofacial abnormalities, primarily midfacial and/or mandibular hypoplasia, which, combined with relative or true macroglossia and short palate, allow for progression to narrow airways and finally to OSA [30,31,32]. However, we did not find any significant association between OSA and malocclusion features; this can be merely explained by the current small sample size. However, the risk of OSA was significantly increased in the presence of posterior cross-bite plus enlarged adenoid and palatine tonsils, confirming, at least in part, what has also been reported in non-DS patients with OSA [33]. Finally, sleep bruxism was reported by 47% of the mothers of our cases. There is indeed a highly variable prevalence of sleep bruxism in DS, but studies are few and not easy to compare; therefore, the interaction between bruxism and OSA in DS children should be more deeply addressed [34]. 

Results from current PG showed that approximately two-thirds of our children with DS have PG-documented OSA, with 62% or 37% prevalence of mild or moderate–severe OSA, respectively. We confirm that the extent of OSA in the DS population is high, although the prevalence in the various studies shows a wide spectrum of distribution due to the different definitions of OSA used [35]. We found that of all PG parameters analysed, only SpO2 mean was inversely related to age and BMI. This is in agreement with other studies that showed that age and BMI had an inverse relationship with SpO2 [36,37]. Several studies have evaluated the relationship between age and AHI, or between BMI and AHI, with still controversial findings [38]. In our study, any age effect on AHI or any correlation between BMI and AHI was found. The broad age range (4–12 years) of our study population could explain these data, confirming previous studies [37,39,40]. In general, clinicians should investigate children with DS for OSA, regardless of age and weight. 

The comparative data from overnight sleep PG parameters to parents’ reports of CSHQ-IT, as well as to the ENT and orthodontic assessment, need further comments. The absence of a correlation between the parental CSHQ-IT report and overnight PG confirms previous reports [24]. When we evaluated the different subscales of the CSHQ-IT, we found an inverse correlation between the score of the subscale exploring “sleep-disordered breathing” and mean SpO2 (r = −0.35; *p* = 0.016). Most of the parents of healthy children with OSA-proved through polysomnography reported correctly snoring and breathing difficulty during sleep [41]. Conversely, DS children with OSA may not always show typical symptoms [42], i.e. “the child snores loudly, stops breathing, snorts and gasps” (items of “sleep-disordered breathing”). These symptoms could be more evident in patients with moderate–severe OSA, who have lower values of mean SpO2. A direct correlation was found between the score of the subscale exploring “sleep duration” and the percentage of the total recording time spent with SpO2 below 90% (r = 0.33; *p* = 0.022). Two items of this subscale were reverse-scored, so the higher scores consistently indicated sleep problems. Therefore, a high T90 can result in “sleep duration” disorder. 

The current combined evidence of a high incidence of OSA at PG and the inaccuracy of parent reports underlines the importance of objective methods to investigate DS sleep. By dividing our study population into children with mild or moderate–severe OSA and children without OSA, we did not find any statistically significant difference in CSHQ-IT total or subscale scores and also in both the ENT and orthodontic findings. In our study, only nocturnal oximetry parameters appeared to be effective for identifying children with DS at risk of OSA, confirming that nocturnal oximetry is a valuable tool for recognizing DS children to be sent promptly to a paediatric sleep laboratory [43,44].

Current ENT analysis revealed that the finding of markedly enlarged adenoid tonsils was associated with higher AHI and T90 at overnight PG, confirming that adenoid hypertrophy is a major determinant of obstruction during sleep in DS subjects [45]. It has been repeatedly reported that ENT manifestations associated with other cofactors significantly contribute to DS airway collapse during sleep [46]. If enlarged adenoid or palatine tonsils or both are found, a surgical intervention (adenoidectomy, tonsillectomy, or adenotonsillectomy) can be proposed, despite residual obstructive symptoms having been reported [47]. 

Although in our population no significant correlation was found between PG and orthodontic data, or between PG and the isolated ENT palatine tonsil size, AHI and ODI were higher in children with associated cross-bite and adenoid and palatine tonsil hypertrophy than in those without. This suggests that in a child with DS, the simultaneous presence of adenotonsillar enlargement and cross-bite further reduces the upper airway space and increases the risk of OSA more than when the isolated dental abnormality is found. Paediatric ENT and orthodontic specialists are usually the first to be consulted to recognize OSA in children with DS. They must be aware of the associated abnormalities and refer patients to sleep experts. 

Our study has both strengths and limitations. First, novel information on a coordinated multidisciplinary approach model to OSA in DS was provided that was lacking in the literature. We designed a prospective, observational study, and all records were collected over a short hospital stay, with benefits to patients and families and less expenses to the national health system. However, the study had some limitations. First, the DS children’ sample from a single centre in Southern Italy was small; therefore, the lack of significant correlations between some variables could be due to the limited sample size. Designing a study of a large and homogeneous DS paediatric population undergoing multiple assessments of sleep problems is not easy. However, our data should hopefully be replicated by multicentre studies including more patients from countries with even different healthcare organizations. Second, respiratory polygraphy rather than PSG to define OSA might have underestimated hypopnoea associated with arousal, but not with oxyhaemoglobin desaturation. However, the choice of respiratory polygraphy was pragmatic, reflecting a trade-off between optimal technology use and compliance in young children with intellectual disabilities like DS [48]. Finally, current data have been collected at a single time point, and no longitudinal findings could be used for comparison; therefore, the authors are not able to comment on the progression of the abnormalities and disease course. 

In conclusion, to the best of our knowledge, this is the first study providing a comparison of combined data of sleep questionnaire, PG, ENT, and orthodontic data from a cohort of children with DS. Our data confirm the high prevalence of OSA in DS children and emphasize the need for using objective measures such as PG for documenting DS sleep problems since no correlation was found between parental reports of sleep problems and sleep study-documented OSA. Oximetry parameters such as ODI, mean SpO2, and T90 appear to be effective in discriminating children with DS at risk of OSA. In other words, when PG or PSG is not available, nocturnal oximetry is a valuable tool to identify DS children to be sent promptly to a paediatric sleep laboratory for the early diagnosis and treatment of OSA. Finally, OSA appears to be correlated with ENT and oral–dental comorbidities, so a coordinated multidisciplinary approach is recommended for DS children. Our results, while extending previous work in DS, will be helpful either to highlight the positive aspects or to identify the potential shortfalls of the multidisciplinary approach to DS, with the final aim of improving patients’ care. Similar multicentre, longitudinal studies would increase the understanding of OSA, a comorbidity that can significantly impact several aspects of the DS course.

## Figures and Tables

**Figure 1 jpm-13-00071-f001:**
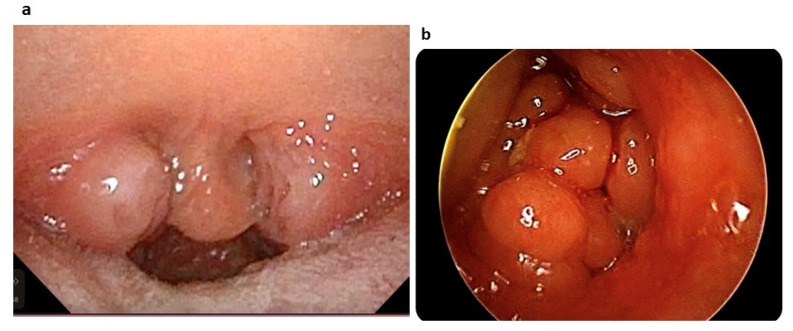
(**a**) Palatine tonsil hypertrophy graded as 3 according to Øverland et al. [14]. (**b**) Adenoid tonsil hypertrophy graded as Grade IV according to Cassano et al. [15].

**Figure 2 jpm-13-00071-f002:**
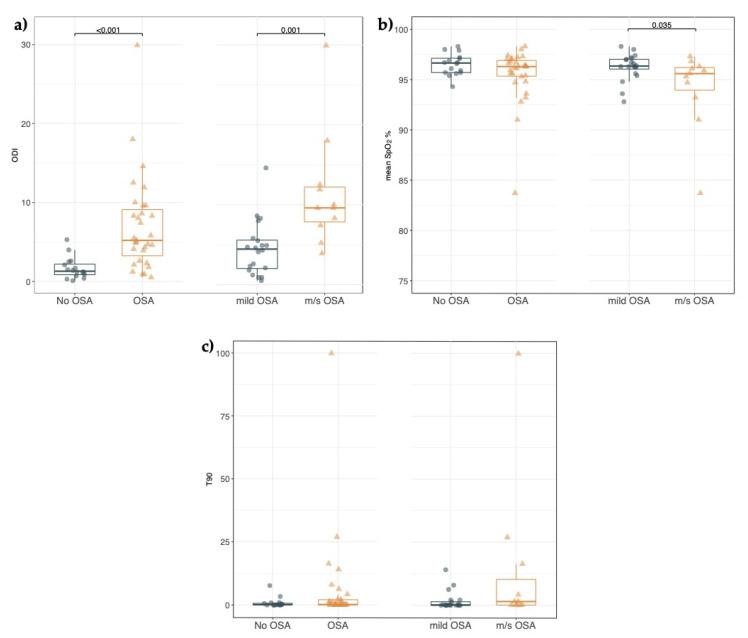
(**a**–**c**). Oximetry indices from nocturnal polygraphy in children with Down syndrome. (**a**) Each point represents the Oxygen desaturation index (ODI) value in children without OSA (non-OSA), with OSA (OSA), with mild OSA (mild-OSA), and moderate to severe OSA (m/s-OSA). Horizontal bars indicate the median values. The Mann–Whitney test was performed. (**b**) Each point represents the mean oxygen saturation value by pulse oximetry (mean SpO2%) in children without OSA (non-OSA), with OSA (OSA), with mild OSA (mild-OSA), and moderate to severe OSA (m/s-OSA). Horizontal bars indicate the median values. The Mann–Whitney test was performed. (**c**) Each point represents the percentage of the total recording time spent with SpO2 below 90% (T90) in children without OSA (non-OSA), with OSA (OSA), with mild OSA (mild-OSA), and moderate to severe OSA (m/s-OSA). Horizontal bars indicate the median values. The Mann–Whitney test was performed.

**Table 1 jpm-13-00071-t001:** Total and subscale scores of CSHQ-IT from mothers of 48 Down syndrome children.

Total Score	63 ± 5.1
Bedtime resistance (items 1-3-4-5-6-8)	11.3 ± 3.3
Sleep onset delay (item 2)	1.4 ± 0.7
Sleep anxiety (items 5-7-8-21)	7.3 ± 2.2
Sleep duration (items 9-10-11)	4.6 ± 1.6
Night wakings (items 16-24-25)	4.8 ± 1.6
Parasomnia (items 12-13-14-15-17-22-23)	9.8 ± 1.8
Sleep-disordered breathing (items 18-19-20)	4.4 ± 1.7
Daytime sleepiness (items 26-27-28-29-30-31-32-33)	14.5 ± 3.5

Data are expressed as mean ± standard deviation. Abbreviations: CSHQ-IT, Children’s Sleep Habits Questionnaire in Italian [12].

**Table 2 jpm-13-00071-t002:** Summary of the results of the ear, nose, throat assessment in Down syndrome children.

		n	%
**Palatine tonsil hypertrophy size**	Size 1	3	9
	Size 2	10	29
	Size 3	14	41
	Size 4	7	21
**Adenoid tonsil hypertrophy size**	grade I	5	15
	grade II	14	41
	grade III	15	44
	grade IV	0	0
**Chronic rhinosinusitis score**	0	1	3
	2	4	12
	4	11	32
	6	9	26
	8	9	26
**Nasal turbinate hypertrophy**	Present	23	68
	Absent	11	32
**Nasal septum deviation**	Present	17	50
	Absent	17	50

**Table 3 jpm-13-00071-t003:** Summary of the results from orthodontic assessment in Down syndrome children.

		n	%
**Facial profile**	Orthognathic	32	68
	Concave	10	21
	Convex	5	11
**Molar class relationship**	I	24	63
	II	6	16
	III	8	21
	NA *	9	-
**Overbite**	Normal	38	81
	Open bite	8	17
	Deep bite	1	2
**Cross-bite**		24	51
**Mother-reported sleep bruxism**		22	47
**Oral parafunctional behaviours**		14	30

Abbreviations: * NA, not available as first molars were not present.

**Table 4 jpm-13-00071-t004:** Overnight polygraphy data from 48 Down syndrome children.

**Total sleep time (min)**	480
	(478; 480)
**Obstructive apnea index**	0
	(0; 1.5)
**Central apnea index**	0
	(0; 0)
**Hypopnea index**	1
	(0; 2.75)
**Apnea-hypopnea index**	2.2
	(0.9; 5.1)
**Oxygen desaturation index**	4
	(1.3; 7.6)
**Mean SpO2 saturation (%)**	96.3
	(95.6; 97)
**T90**	0.2
	(0; 1.5)

Data are expressed as the median and interquartile range (in parentheses). Abbreviations: SpO2, oxygen saturation by pulse oximetry; T90, percentage of the total recording time spent with SpO2 below 90%.

**Table 5 jpm-13-00071-t005:** Correlations between overnight polygraphy and CSHQ-IT data in the Down syndrome study cohort.

	Total Score	Bedtime Resistance	Sleep Onset Delay	Sleep Duration	Sleep Anxiety	Night Wakings	Parasomnias	Sleep-Disordered Breathing	Daytime Sleepiness
**AHI**	0.07	0.04	−0.05	0.09	0.01	−0.04	−0.13	0.16	0.12
**ODI**	−0.01	0.13	−0.18	0.09	−0.01	−0.14	−0.28	0.04	0.13
**Mean SpO_2_**	−0.17	0.05	0.02	−0.22	0.14	−0.18	0.03	−0.35 *	−0.14
**T90**	0.11	−0.03	−0.04	0.33 *	0.01	0.23	−0.06	0.19	−0.07

Results are expressed as Spearman correlation coefficient (r). * *p* < 0.05. Abbreviations: CSHQ-IT, Children’s Sleep Habits Questionnaire adapted to the Italian language [12]; AHI, apnoea–hypopnoea index; ODI, oxygen desaturation index; SpO2, oxygen saturation by pulse oximetry; T90, percentage of the total recording time spent with SpO2 below 90%.

**Table 6 jpm-13-00071-t006:** Correlation between ear, nose, and throat data and major overnight polygraphy results in children with Down syndrome.

	Adenoid Tonsils Hypertrophy Size			Palatine Tonsils Hypertrophy Size			Nasal Turbinate Hypertrophy			Nasal Septum Deviation		
	I/II Grade	III Grade	*p* *	I/II Grade	III/IV Grade	*p* *	Present	Absent	*p* *	Present	Absent	*p* *
**AHI**	1 (0.6; 4.5)	5 (0.9; 8.1)	0.04	1.5 (0.6; 5.5)	2.9 (0.9; 7.1)	0.36	2.1 (0.9; 5.5)	2.2 (0.8; 6)	1	2.4 (1; 6.2)	1.5 (0.8; 5.5)	0.62
**ODI**	4 (1.8; 5.7)	8 (1.7; 9.6)	0.08	4.1 (2.3; 6.7)	4.6 (1.4; 8.9)	0.92	4.7 (1.7; 8.6)	2.6 (1.5; 8.3)	0.60	4.9 (2; 8.8)	2.6 (1.6; 8.3)	0.59
**Mean SpO_2_**	96.3 (95.5; 97)	96.3 (94.8; 97.1)	0.58	96.1 (95; 96.8)	96.7 (95.7; 97.1)	0.27	96.3 (95.6; 97.1)	96.7 (94.3; 97)	0.71	96.3 (95.7; 96.8)	96.7 (94.5; 97.1)	0.62
**T90**	0.1 (0; 0.2)	1.2 (0.1; 3.4)	0.01	0.2 (0; 1.8)	0.1 (0; 1.8)	0.75	0.1 (0; 1.5)	0.3 (0; 7.7)	0.46	0.1 (0; 1.4)	0.4 (0; 4.9)	0.12

Data are expressed as median and interquartile range (in parenthesis). The Mann-Whitney test was performed. * *p* < 0.05. Abbreviations: AHI, Apnea-hypopnea index; ODI, Oxygen desaturation index; SpO2, oxygen saturation by pulse oximetry; T90, % of the total recording time spent with SpO2 below 90%.

**Table 7 jpm-13-00071-t007:** Comparison of polygraphy parameters between group A * and group B # patients with Down syndrome.

	Group A * (n = 6; 15.8%)	Group B # (n = 32; 74.2%)	*p* **
**Apnea-hypopnea index**	7.1 (4.3; 9.4)	1.4 (0.8; 4.6)	0.01
**Oxygen desaturation index**	8.9 (4.6; 10.2)	3.3 (1.2; 7.1)	0.04
**Mean SpO_2_**	96.3 (91.9; 97.3)	96.3 (95.6; 96.9)	0.98
**T90**	0.1 (0; 26.1)	0.2 (0; 2)	0.76

* Group A: Patients with cross-bites, grade III adenoid tonsil hypertrophy, and grade III–IV palatine tonsil hypertrophy. # Group B: Patients without cross-bite grade III adenoid tonsil hypertrophy, and grade III–IV palatine tonsil hypertrophy. Data are expressed as median and interquartile range (in parentheses). The Mann–Whitney test was performed. ** *p* < 0.05. Abbreviations: SpO2, oxygen saturation by pulse oximetry; T90, percentage of the total recording time spent with SpO2 below 90%.

## Data Availability

Not applicable.

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
