# Peer review of "Obstructive Sleep Apnoea in Children with Down Syndrome: A Multidisciplinary Approach"

_jpm, 2022, doi:10.3390/jpm13010071_

Round 1
Reviewer 1 Report
Thank you for giving me the opportunity to review this interesting article
Even if English is not my native language, I would suggest to review some few writing mistakes
I have to ask only “minor revision”
Abstract
We sought to evaluate the effect of a multidisciplinary approach to DS OSA.
I would suggest …approach to DS with OSA.
Enlarged palatine or adenoid tonsils and chronic rhinosinusitis were found in 62%, 85% and 85% of cases.
The understanding of this sentence is not immediate. I would suggest to make it more clearly.
A coordinated multidisciplinary approach with overnight oximetry is a valuable tool when developing diagnostic protocols for OSA in DS.
This sentence is not completely clear and supported by your data. You did not performed overnight oximetry but polygraphy, but you conclude with oximetry. I understand that you find interesting results on the oximetry metrics, but I feel the sentence confusing.
Materials and Methods
I would suggest to the authors to include in this section the reasons why they choose 4 to 12 years old patients.
Discussion
As first sentence into this section, I would suggest to include a short paragraph in which the authors tell us briefly:
What have they done
In which patients
What important results they achieved
Please consider the same as above for the conclusion. I think that here you must clearly refers on the most important result obtained in the study
Only nocturnal oximetry appeared to be effective for identifying children with DS at risk of OSA, confirming that nocturnal oximetry is a valuable tool for recognising DS children to be sent promptly to a paediatric sleep laboratory [44, 45].
This sentence is confusing because there is a mix of polygraphy, oximetry, your own data and data from literature. I would suggest to change in “In our study, only nocturnal oximetry parameters appeared to be ….”.
Author Response
Thank you for giving me the opportunity to review this interesting article
Even if English is not my native language, I would suggest to review some few writing mistakes.
I have to ask only “minor revision”
REPLY: We thank the Reviewer for his/her comments and suggestions, which certainly improves the quality of our study.
The current version of our manuscript has been modified according to his/her comments.
We have reread the manuscript with a native speaker teacher and we have made some corrections.
The modifications performed are highlighted in yellow.
Please find below a detailed list of the performed changes:
Abstract
We sought to evaluate the effect of a multidisciplinary approach to DS OSA.
I would suggest …approach to DS with OSA.
REPLY: Thanks for the observation. We added “with” (page 1, line 23).
Enlarged palatine or adenoid tonsils and chronic rhinosinusitis were found in 62%, 85% and 85% of cases.
The understanding of this sentence is not immediate. I would suggest to make it more clearly.
REPLY: We have rewritten the sentence (page 1, lines 27-28).
A coordinated multidisciplinary approach with overnight oximetry is a valuable tool when developing diagnostic protocols for OSA in DS.
This sentence is not completely clear and supported by your data. You did not performed overnight oximetry but polygraphy, but you conclude with oximetry. I understand that you find interesting results on the oximetry metrics, but I feel the sentence confusing.
REPLY: We fully agree with the reviewer. We have replaced “overnight oximetry” with “overnight polygraphy” (page 2, line 39).
Materials and Methods
I would suggest to the authors to include in this section the reasons why they choose 4 to 12 years old patients.
REPLY: We added the reasons for choosing DS children aged 4-12 years (page 2; lines 81-82).
Discussion
As first sentence into this section, I would suggest to include a short paragraph in which the authors tell us briefly:
What have they done
In which patients
What important results they achieved
Please consider the same as above for the conclusion. I think that here you must clearly refers on the most important result obtained in the study
REPLY: We thank the Reviewer for raising the question. In discussion both at the beginning (page 13, lines 368-376) and at the end (page 15, lines 498-509), we inserted a short paragraph summarizing the following points: “What have they done - In which patients - What important results they achieved”.
Only nocturnal oximetry appeared to be effective for identifying children with DS at risk of OSA, confirming that nocturnal oximetry is a valuable tool for recognising DS children to be sent promptly to a paediatric sleep laboratory [44, 45].
This sentence is confusing because there is a mix of polygraphy, oximetry, your own data and data from literature. I would suggest to change in “In our study, only nocturnal oximetry parameters appeared to be ….”.
REPLY: we now modified as suggested (page 14; line 457).
Reviewer 2 Report
Comments to the authors
Thank you for letting me review the article entitled “Obstructive sleep apnea in children with Down syndrome: a multidisciplinary approach”.
The manuscript presents a multidisciplinary approach to the diagnosis of OSA in Down Syndrome’s children. I think that the most interesting finding is the fact that despite a poor sleep as reported by the mother and assessed by the questionnaire, the PG did not confirm such data.
More than novel in its findings and research question, the manuscript is very well written in the methodology part.
To improve the quality of the study, I only have two major corrections and several minor requests. See below.
Major correction:
- was a sample size calculation performed? If it was not, the authors can provide a power analysis
- the paragraph in lines 232-234 where the group of healthy subjects is presented need to be clarified in advance. There is no hint from the paper so far that a health group will be used for comparison, and it cannot be surprise in the results section. I encourage the authors to state it in advance, in the statistical analysis section. Also, is this healthy group taken as comparison age- and BMI-matched to the study group? If this is not the case, the comparison may result inappropriate. And “healthy children” means without DS or without OSA? Please provide all this additional information.
Minor correction:
- line 102: “as” instead of “a previously described”
- It would be nice and informative if the two classifications of the description of palatine tonsillar hypertrophy (paragraph in lines 120-123) and adenoid tonsils hypertrophy (paragraph lines 126-133) were supported by pictures or drawing. If it is possible, I would encourage the authors to do so.
- how can the orthodontist evaluate the presence of sleep bruxism (line 157)? I agree with the endorsement of self-reported or observed oral parafunctional behaviors. I would remove the sleep bruxism, unless specify that it is reported by the parents. Sleep bruxism cannot be observed from an oral examination. Were the episodes of sleep bruxism confirmed with the overnight PG?
- in the statistical analysis, it is not clear which two groups were compared. The authors have to first specify that two groups were created on the basis of diagnosis of OSA, then the analysis follows.

Author Response
Thank you for letting me review the article entitled “Obstructive sleep apnea in children with Down syndrome: a multidisciplinary approach”.
The manuscript presents a multidisciplinary approach to the diagnosis of OSA in Down Syndrome’s children. I think that the most interesting finding is the fact that despite a poor sleep as reported by the mother and assessed by the questionnaire, the PG did not confirm such data.
More than novel in its findings and research question, the manuscript is very well written in the methodology part.
To improve the quality of the study, I only have two major corrections and several minor requests. See below.
REPLY: We thank the Reviewer for his/her comments and suggestions, which certainly improves the quality of our study.
The current version of our manuscript has been modified according to his/her comments.
The modifications performed are highlighted in yellow.
Please find below a detailed list of the performed changes:
Major correction:
- was a sample size calculation performed? If it was not, the authors can provide a power analysis.
REPLY: We thank the reviewer for his/her suggestion. Our study had an exploratory objective, and
we did not have any a-priori hypothesis that could have guided a formal power analysis. The sample size was decided on a convenience basis. We fully agree that our findings could be biased towards false negative results due to the small sample size and we recognized this as one of the main limits of this study; however, as recommended by several authors from statistician community (Heckman MG, Davis JM 3rd, Crowson CS. Post Hoc Power Calculations: An Inappropriate Method for Interpreting the Findings of a Research Study. J Rheumatol. 2022; 49:867-870. - Heinsberg LW, Weeks DE. Post hoc power is not informative. Genet Epidemiol. 2022; 46:390-394), we preferred not to perform a post-hoc power analysis as these calculations could be misleading and not informative for data interpretation.
- the paragraph in lines 232-234 where the group of healthy subjects is presented need to be clarified in advance. There is no hint from the paper so far that a health group will be used for comparison, and it cannot be surprise in the results section. I encourage the authors to state it in advance, in the statistical analysis section. Also, is this healthy group taken as comparison age- and BMI-matched to the study group? If this is not the case, the comparison may result inappropriate. And “healthy children” means without DS or without OSA? Please provide all this additional information.
REPLY: We thank the Reviewer for this insightful observation. We have now specified that about CSHQ-IT data we compared DS children with children without DS. Two groups were matched for age, sex and BMI (page 5, lines 201-203).
Minor correction:
- line 102: “as” instead of “a previously described”
REPLY: we have corrected it.
- It would be nice and informative if the two classifications of the description of palatine tonsillar hypertrophy (paragraph in lines 120-123) and adenoid tonsils hypertrophy (paragraph lines 126-133) were supported by pictures or drawing. If it is possible, I would encourage the authors to do so.
REPLY: According to this useful comment, we now added two pictures (figure 1a-1b): figure 1a, about palatine tonsillar hypertrophy; figure 1 b, about adenoid tonsils hypertrophy.
Following these changes, the previously reported figure 1 (a-c) has been reported as figure 2 (a-c).
- how can the orthodontist evaluate the presence of sleep bruxism (line 157)? I agree with the endorsement of self-reported or observed oral parafunctional behaviors. I would remove the sleep bruxism, unless specify that it is reported by the parents. Sleep bruxism cannot be observed from an oral examination. Were the episodes of sleep bruxism confirmed with the overnight PG?
REPLY: We agree with these observations on sleep bruxism. In our analysis, the specialist in orthodontics asked mothers about sleep bruxism. We have specified this in the revised manuscript (page 4, lines 168-170, page 13, lines 420-421). However, we could not evaluate sleep bruxism by overnight poligraphy.
- in the statistical analysis, it is not clear which two groups were compared. The authors have to first specify that two groups were created on the basis of diagnosis of OSA, then the analysis follows.
REPLY: We really appreciate this remarks that allow us to better explain the statistical analysis plan. Actually, due to the presence of several between-group comparisons, the expression used in statistical analysis section referred to any comparison involving two groups and that required the statistical assessment of the significance of the observed difference. We rephrased the sentence in the corresponding paragraph in order to be more precise (page 5, lines 198-203 ).